# Pro12Ala PPAR-γ2 and +294T/C PPAR-δ Polymorphisms and Association with Metabolic Traits in Teenagers from Northern Mexico

**DOI:** 10.3390/genes11070776

**Published:** 2020-07-10

**Authors:** Martín A. Carrillo-Venzor, Nancy R. Erives-Anchondo, Janette G. Moreno-González, Verónica Moreno-Brito, Angel Licón-Trillo, Everardo González-Rodríguez, Pilar del Carmen Hernández-Rodríguez, Sandra A. Reza-López, Verónica Loera-Castañeda, Irene Leal-Berumen

**Affiliations:** 1Faculty of Medicine and Biomedical Sciences, Autonomous University of Chihuahua, Circuito Universitario, Campus II, Chihuahua 31109, Mexico; a274058@uach.mx (M.A.C.-V.); smtp_gmail@uach.mx (N.R.E.-A.); jgmoreno@uach.mx (J.G.M.-G.); vmoreno@uach.mx (V.M.-B.); alicon@uach.mx (A.L.T.); evegonzal@uach.mx (E.G.-R.); sreza@uach.mx (S.A.R.-L.); 2Faculty of Chemical Sciences, Autonomous University of Chihuahua, Circuito Universitario, Campus II, Chihuahua 31109, Mexico; pilar_hernandez@inclar.com; 3CIIDIR-IPN, Durango 34220, Mexico; veronica.loera@gmail.com

**Keywords:** Pro12Ala, +294T/C, PPARs polymorphisms, metabolic traits, allele frequencies, major/minor allele, genotypes

## Abstract

Peroxisome proliferator-activated receptors (PPARs) play roles in glucose and lipid metabolism regulation. Pro12Ala PPAR-γ2 and +294T/C PPAR-δ have been associated with dyslipidemia, hyperglycemia and high body mass index (BMI). We compared metabolic traits and determined associations with Pro12Ala PPAR-γ2 or +294T/C PPAR-δ polymorphism among teenagers from different ethnicity. Four hundred and twelve samples with previous biochemical and biometric measurements were used. Genomic DNA from peripheral blood was extracted and analyzed by end-point PCR for Pro12Ala PPAR-γ2. The +294T/C PPAR-δ PCR product was also digested with Bsl I. Two genotype groups were formed: major allele homozygous and minor allele carriers. Pro12Ala PPAR-γ2 G minor allele frequencies were: 10% in Mestizo-1, 19% in Mestizo-2, 23% in Tarahumara, 12% in Mennonite, and 17% in the total studied population. The +294T/C PPAR-δ C minor allele frequencies were: 18% in Mestizo-1, 20% in Mestizo-2, 6% in Tarahumara, 13% in Mennonite, and 12% in the total studied population. Teenagers with PPAR-γ2 G allele showed a greater risk for either high waist/height ratio or low high-density lipoprotein; and, also had lower total cholesterol. Whereas, PPAR-γ2 G allele showed lower overweight/obesity phenotype (BMI Z-score) frequency, PPAR-δ C allele was a risk factor for it. Metabolic traits were associated with both PPAR polymorphisms.

## 1. Introduction

The worldwide prevalence of obesity in 2015 was 5% in children and 12% in adults [1]. According to the Organization for Economic Co-operation and Development (OECD) Heavy Burden of Obesity 2018, close to 60% of the population from the 36 countries members of the OECD have overweight and nearly 25% have obesity. Obesity and related diseases will reduce life expectancy 0.9–4.2 years in the future and 92 million premature deaths are expected due to obesity-related diseases by 2050 [2]. In the US, the obesity rate has increased from 10.5 to 20.6% in 12–19 year-old teenagers between 1988 and 2014 [3]. Obesity has also increased drastically in Mexico. According to the Encuesta Nacional de Salud y Nutrición 2018, the overweight/obesity prevalence in Mexico was 38.4% for 12–19 year-olds [4]. 

PPARs play roles in physiology, pathology, and whole-body energy regulation, lipid and glucose metabolism. Pro12Ala PPAR-γ2 and +294T/C PPAR-δ have been associated with dyslipidemia, hyperglycemia, and overweight/obesity. PPARs are peroxisome proliferator-activated receptors and members of the nuclear hormone receptor superfamily, located in a wide variety of tissues and cells. They are ligand-activated nuclear transcription factors that interact with complex metabolic networks [5,6]. The PPAR-γ gene is located on chromosome 3p25 (OMIM number 601487) and it encodes three proteins isoforms [7,8]. The PPAR-γ1 isoform is expressed in most tissues; the γ3 is seen in macrophages, colon epithelium, and adipose tissue [9]. However, the γ2 is specific for adipose tissue and it can be activated by fatty acids including prostanoids, thiazolidinediones, and other insulin-sensitizing antidiabetic agents [10,11]. The Pro12Ala polymorphism is a missense coding variant (CCA to GCA) in γ2, which results in a proline (major allele) to alanine (minor allele) substitution (rs1801282) [12]. The presence of this single nucleotide polymorphism (SNP) has been associated with type 2 diabetes (T2D) related traits, hypertension, abnormal lipid profile, insulin resistance, and greater body mass index (BMI) [9,13]. Several meta-analysis studies have reported between 13–16% increased risk of T2D in adult population from Europe, North America, and East Asia with a Pro allele [8,14]. However, different results have been reported regarding BMI and Pro12Ala polymorphism in other ethnic groups [15,16,17,18,19]. The prevalence of the modified Ala12 allele varies from about 4% in Asian populations to 28% in Caucasians [20].

The PPAR-δ gene is located in chromosome 6p21.3 (OMIM number 600409) with 13 exons [21]. It has ubiquitous expression; however, its greatest expression occurs in tissues with high lipid metabolism, including the small intestine, heart, adipose tissue, and skeletal muscle [22]. PPAR-δ can be activated by polyunsaturated fatty acids and synthetic compound GW501516 [23]. The +294T/C polymorphism (rs2016520) is also called −87 T/C or +15C/T and is a T (major allele)–C (minor allele) base exchange in the 5′-UTR region of the gene [24,25]. The presence of this SNP has been related to several metabolic traits. It is mostly associated with alterations in the blood levels of HDL, LDL, VLDL, and triglycerides [24,25,26,27,28]. There is also an association between the C allele and BMI, glucose levels, height, and coronary heart disease (CHD) [24,26,27,29,30,31,32,33,34]. The prevalence of the C allele varies among different populations: 22.5% to 31.04% in Chinese people, 19.2% in Germans, 20.1% in French Canadians and 26% in Mexicans [22,25,28,35,36].

A few studies in Mexican teenagers have shown association with Pro12Ala PPAR-γ2 polymorphism with obesity-related traits. Stryjecki et al. reported an association between this polymorphism with insulin resistance in Mexican children [11]. No studies about +294T/C PPAR δ polymorphism and metabolic traits in Mexican children were found. The aim of this study is to determine the association between Pro12Ala PPAR-γ2 or +294T/C PPAR-δ polymorphisms with metabolic traits in Mexican teenagers from different ethnic populations in Chihuahua, Mexico. We focused on teenagers to investigate whether polymorphisms and clinical metabolic traits relate, since the early detection of inherited polymorphisms, which cannot be modified, will allow us to promote preventive behaviors as eating healthy food, eating schedules, and physical activity.

## 2. Materials and Methods

### 2.1. Study Population

The Facultad de Medicina y Ciencias Biomédicas, Ethical Research Committee Board (Protocol registration numbers FM-FM-A269/12 and CI-040-17) from the Universidad Autónoma de Chihuahua approved the study protocol; written authorization was obtained from one of the parents or tutors. There were 412 blood samples collected from unrelated adolescents aged 12–18 years. Samples were collected from a previous scholar health care study in three different populations (Mestizo, Tarahumara, and Mennonite) from Chihuahua, Mexico. However, two Mestizo groups were considered (Mestizo-1 and Mestizo-2), since they were from different geographic locations. Moreover, we found significant differences when clinical metabolic traits were compared between them. On the other hand, a parallel study in our lab (same mestizos) showed significant differences in mitochondrial haplotype frequencies between these two groups (results not shown). Recruitment was done in collaboration with public high schools not randomly selected. The data used to support the findings of this study are restricted by the Ethics Board “Comité de Ética en Investigación de la Facultad de Medicina y Ciencias Biomédicas de la Universidad Autónoma de Chihuahua”, in order to protect participants’ confidentiality. Data are available from Irene Leal-Berumen, ileal@uach.mx, for the researchers who meet the criteria for access to confidential data.

### 2.2. Phenotyping

All of the participants were weighed while using a digital scale (Tanita BC-418, Ilinois, USA), and height was measured with a portable stadiometer (HM200P, Taichung, Taiwan). The waist circumference (WC) was measured at the midpoint between the lowest rib and the iliac crest after a normal exhalation with students in the standing position, and percentile waist circumference was classified as <p90 normal or ≥p90 increased [37,38]. The waist-to-height ratio (WHR) is a novel abdominal overweight indicator < 0.45 is normal, ≥0.45 is overweight, and ≥0.5 is obese [39]. Body mass index was calculated as weight (kg)/height(m^2^) and individual values were then compared to BMI-for age and sex-specific reference charts in order to obtain their percentile and the Z-score value [40,41,42]. The percentile BMI (pBMI) was classified as p5 to <p85 as normal, p85 to <p95 overweight, and ≥p95 obese according to CDC growth reference charts, while for Z-score > 1.04 [40]. Blood samples were obtained after 8–12 h fast to measure the following via an automated clinical analyzer (Prestige 24i; Tokio, Boeki Medical System LTD, Japan): fasting glucose, total cholesterol (TC), high-density lipoprotein cholesterol (HDL), low-density lipoprotein cholesterol (LDL), and triglycerides (TG). The metabolic traits cut-off points included high blood pressure, defined as systolic and/or diastolic blood pressure ≥ p90 (SBP/DBP); waist circumference ≥ p90; TG ≥ 110 mg/dL; HDL ≤40 mg/dL; and, high blood glucose ≥ 110 mg/dL according to Cook et al. [37]. Other metabolic traits risk values were included TC ≥ 150 mg/dL, LDL ≥ 110 mg/dL, and the atherogenic index (AI = TC/HDL; male > 4.0 and female > 3.5) [43,44]. 

### 2.3. Genotyping

Genomic DNA was isolated from a peripheral blood buffy coat while using a Master Pure Epicentre kit (Thermo Scientific, Madison, WI, USA). The Pro12Ala PPAR-γ2 polymorphism was genotyped by mutagenically separated PCR with two different length allele-specific downstream primers (P1 and P2) and a common upstream primer (P3). The P1 (5′-GTGTATCAGTGAAGGAATCGCTTTCT**T**G-3′) was specific for the C allele (Pro); and P2 (5′-TTGTGATATGTTTGCAGACA**AG**GTATCAGTGAAGGAATCGCTTTGTGC-3′) bound to the G allele (Ala). The P3 (upstream primer) was 5′-TTTCTGTGTTTATTCCCATCTCTCCC-3′. The bases underlined and in bold type indicate the location of mismatches to maintain the specificity of the two separate amplification reactions. Here, DNA (30 ng) was added to a 25 mL reaction mix containing 3 mM MgCl_2_, 5 pmol P1, 5 pmol P2, 5 pmol P3, 200mM dNTP´s, and 1 U Taq polymerase (Invitrogen, Carlsbad, CA, USA). The PCR conditions used an initial denaturation of 3 min. at 94 °C followed by 35 cycles of denaturation at 94 °C for 45 s, annealing at 62 °C for 45 s, and extension at 72 °C for 45 s. The final extension step was 5 min. at 72 °C (Agilent SureCycler 8800, Santa Clara, CA, USA). A 230 bp product identified the Pro allele, and the Ala-specific product was 250 bp. Electrophoresis used 3.5% agarose (Appendix A) [9]. The +294T/C PPAR-δ polymorphism genotyping used endpoint PCR with 5′-CATGGTATAGCACTGCAGGAA-3′ (forward) and 5′-CTTCCTCCTGTGGCTGCTC-3′ (reverse) primers. The 25-mL mix reaction contained 1.5 mM MgCl_2_, 0.2 μM forward primer, 0.2 μM reverse primer, 200 μM dNTPs, and 1 U Taq polymerase (Invitrogen, Carlsbad, CA, USA). The PCR conditions were the same as in the Pro12Ala polymorphism. The 269 bp resulting PCR product was digested with 5 U fast Bsl I (Thermo Scientific, Madison, WI, USA) restriction enzyme for 1 h at 37 °C. The 12% polyacrylamide electrophoresis was used to identify the three different genotypes: single 269 bp fragment for TT, three fragments for TC (269, 167, and 102 bp), and two fragments for CC (167 and 102 bp) (Appendix A) [25].

### 2.4. Statistical Analysis

An exploratory analysis was performed to verify the data quality and observe variable distribution and frequencies. Normality tests were conducted for variables measured in ratio scale. The mean and standard deviation or median and interquartile range were used to describe normally (SBP, glucose, total cholesterol, and LDL) or non-normally distributed variables (the other metabolic traits), respectively. Metabolic traits measured in ratio scale were analyzed as continuous or categorical variables with the cutoffs defined in Section 2.2. For comparisons among ethnic groups or polymorphism groups, we used either ANOVA or t-test for normally distributed variables, and Kruskal–Wallis or Wilcoxon rank sum test (non-parametric tests) for non-normally distributed variables. For linear regression analyses we log-transformed non-normally distributed variables, as described in the following paragraphs.

To compare metabolic traits measured in continuous scale among ethnic groups we used ANOVA—for normally distributed—or Kruskal–Wallis test—for non-normally distributed—variables, followed by post hoc Bonferroni´s test or Dunn´s test with Bonferroni adjustment, respectively, for pairwise comparisons. Significant differences between ethnic groups were indicated by different superscripts (a–e). When groups were not statistically different, they share the same letter; therefore, some groups may have more than one superscript letter. The frequency and percentage were used to describe categorical variables. The Chi^2^ test was used to compare groups and to verify Hardy-Weinberg equilibrium (HWE). 

We formed two groups according to PPAR-γ2 and PPAR-δ genotypes (major vs. minor allele carriers). Comparison of metabolic traits between the two allelic groups was performed with t-test or Wilcoxon´s rank sum test for normal or non-normally distributed variables, respectively. The α value was adjusted for multiple comparisons by the Holm–Bonferroni method, to correct for family-wise error rate.

Multiple linear regression models were used to adjust the relation between metabolic traits (as continuous variables) and genotype groups for potential confounders and effect modifiers using a forward inclusion of variables. Metabolic trait variables with a non-normal distribution were log-transformed (TG, HDL, VLDL, and atherogenic index) to reach normality, only in the case of the diastolic blood pressure, we left the original values, because the distribution did not reach normality with any of the tested transformations. The model residuals were analyzed and a heteroskedasticity test was performed to verify linear regression assumptions. In further analyses, we included ethnic groups as dummy variables and as fixed effects, for the analyses performed in the entire sample. Because we found interaction effects between the polymorphism and ethnic groups, we conducted a stratified analysis, by ethnic group.

Dichotomous variables of metabolic traits were compared by Pearson’s Chi^2^ or Fisher exact test. Logistic regression models were used to calculate the odds ratio (OR) of metabolic traits (dichotomous) according to the genotype groups. The reference group was homozygous for the major allele for both PPAR polymorphisms. Logistic regression analysis was performed on the total studied population only due to the limited sample size in some ethnic groups, which led to very low frequencies in some categories. A *p*-value < 0.05 was considered to be statistically significant, unless otherwise specified. All of the statistical analyses were performed with STATA software (v. 11.0, StataCorp, College Station, TX, USA). 

## 3. Results

### 3.1. Population Characteristics and Metabolic Traits among Them

Three main ethnic populations were studied, Mestizo, Tarahumara, and Mennonite. However, the Mestizo group was separated in two: Mestizo-1 and Mestizo-2, since they were from a different location and showed significant differences between them. The median age in the total studied population was 14 years and 60.92% were females. The Mestizo-2 group had the highest overweight/obese phenotype proportion according to their pBMI (Table 1).

Metabolic traits were compared among the ethnic populations (Table 2). Relevant differences were observed among them. The Mestizo-2 group showed greater values for BMI Z-score, waist circumference, and waist/height ratio, but lower glucose levels when compared to the other populations. The Tarahumara and Mestizo-2 populations had higher triglycerides values compared to Mestizo-1 and Mennonite, whereas the Tarahumara group had the lowest HDL levels with greater atherogenic index as compared to the others. The prevalence of altered metabolic traits in total studied population included overweight/obesity 20.15% (by BMI Z-score), high blood pressure 32.28%, high waist circumference 3.4%, high waist/height ratio 39.56%, hyperglycemia 1.21%, hypertriglyceridemia 25.73%, high cholesterol 12.14%, low HDL 32.11%, high LDL 3.88%, high VLDL 7.04%, and high atherogenic index 31.34%.

### 3.2. Pro12Ala PPAR-γ2 and +294T/C PPAR-δ

The genotype and allelic frequency distribution were compared among populations. For PPAR-γ2, no significant difference was observed among Mestizo-1, Mestizo-2, and Mennonite populations; however, the genotype distribution in Tarahumara was different between Mestizo-1 and Mennonite, but not with Mestizo-2 (Table 3). Similar results were observed for PPAR-δ genotype distribution—only the Tarahumara population showed a different genotype distribution (Table 4). The polymorphisms Pro12Ala PPAR-γ2 and +294T/C PPAR-δ were in Hardy–Weinberg equilibrium in all of the ethnic groups and total studied population.

### 3.3. Metabolic Traits by Pro12Ala PPAR-γ2 and +294T/C PPAR-δ Polymorphism Genotype: Major Allele Homozygous vs. Minor Allele Carriers in the Total Studied Population

Statistical analysis showed significant differences in waist/height ratio and HDL median values between PPAR-γ2 alleles. The α value was adjusted for multiple comparisons by the Holm–Bonferroni method, to correct for family-wise error rate. This association remained significant after adjusting for multiple testing (cut off *p* value = 0.0125), whereas the difference in waist/height ratio did not (cut off *p* value = 0.006). Individuals showed significant differences in TG, HDL, and VLDL median values, according with their PPAR-δ genotype. However, the association between PPAR-δ and HDL did not remain significant after correcting for multiple testing, while significance remained for TG and VLDL (Table 5). 

### 3.4. Association between Pro12Ala PPAR-γ2 and Metabolic Traits

Among the clinical variables, sex was significantly associated (*p* < 0.05) with all metabolic traits, except for diastolic blood pressure and age was also associated with all, except for LDL and atherogenic index. Therefore, we included them as covariates in multivariate models. Associations between metabolic traits and the polymorphism Pro12Ala PPAR-γ2 were found with glucose, HDL, and atherogenic index in the total studied population, adjusting for clinical variables (age, sex, and or waist circumference). Minor allele carries had lower HDL and glucose levels and higher atherogenic index, than major allele homozygous. However, analysis according to ethnic populations only showed a significant association in the Mestizo-1 group with SBP and HDL. The minor allele carriers had ~5.59 mmHg lower SBP (adjusted by waist circumference, sex, and age) and ~0.90 mg/dL lower HDL than major allele homozygotes (log(−0.11), adjusted by waist circumference and sex) (Table 6). After also adjusting for ethnic group, only the association of atherogenic index remained significant. The interaction effect of sex and age for SBP showed that men had a slight elevation in SPB as age increased. The PPAR-γ2 and atherogenic index relation was modified by the waist circumference. Atherogenic index increased as waist circumference increased, but this relation differed by the studied polymorphism (Table 6 footnote). After also adjusting for ethnic group (last column), only the association of atherogenic index remained significant. An interaction effect between the polymorphism and the ethnic group was also observed for LDL.

### 3.5. Association between +294T/C PPAR-δ and Metabolic Traits

Associations between +294T/C PPAR-δ and metabolic traits in the total studied population were observed with glucose, TG, TC, and VLDL. C allele carries had 2.57 mg/dL less glucose (adjusted by waist circumference, sex and age), 0.89 mg/dL less TG (log(−0.12), adjusted by sex and age), 0.89 mg/dL less VLDL (log(−0.12), adjusted by waist circumference, sex and age), and 79.02 mg/dL more TC (adjusted by sex and age) than major allele homozygotes. Among ethnic groups, the only association was found in the Tarahumara population with HDL where C allele carriers had 0.90 mg/dL less HDL (log(−0.10), adjusted by waist circumference, sex, and age) than major allele homozygotes (Table 7). The relation between PPAR-δ allele and HDL was modified by sex. In male teenagers, the HDL levels were higher in C allele carriers versus major allele homozygotes; no differences were noted in females (Figure 1A). In the total studied population, PPAR-δ and LDL relation was also modified by sex. Males with the major allele had the lowest values of LDL versus females and males with minor allele carriers (Figure 1B). The same interaction was found in Mestizo-1 population (Table 7 footnote). After adjusting for ethnic group (Table 7, last column), the association of the polymorphism and glucose levels remained significant, as well as the interaction effect between the polymorphism and sex for LDL. We also observed a significant interaction effect between ethnic group and this polymorphism in its relation to HDL and atherogenic index.

### 3.6. Logistic Regression

In the total studied population, Pro12Ala PPAR-γ2 minor allele carriers had greater likelihood of having an elevated waist-height ratio (OR = 1.79, adjusted by sex) and 2.5-fold greater chance of having low HDL values (adjusted by waist circumference, sex, and age) than major allele homozygous. In contrast, they had a nearly two-fold lower likelihood of having a BMI Z-score suggestive of overweight/obesity (OR = 0.46) and increased TC (OR = 0.47, adjusted by sex and age) than major allele homozygous. The PPAR-δ minor allele carriers had greater likelihood (OR = 1.78) of having a BMI Z-score that was suggestive of overweight/obesity (Table 8). No significant interactions were seen between both minor alleles.

## 4. Discussion

In this study, we analyzed the association between the Pro12Ala PPAR-γ2 or +294T/C PPAR-δ polymorphisms and metabolic traits in teenagers from the North of Mexico, including Mestizo, Tarahumara, and Mennonite populations. Hispanic populations have a higher tendency for obesity than other Caucasians due to their genetic origin and cultural characteristics [45,46,47]. Mexican-origin population have Native American and Spanish admixture. However, Salzano et al. reported that Amerindian ancestry is most prevalent in the general population of Mexico followed by European ancestry [48]. The Mexican territory is extensive, it borders with countries of North and Central America. In addition, Mexico has a great history of Spanish colonization and the arrival of a large number of immigrants of African and European origin. Therefore, it is not surprising to find diversity in the population in this country. The interest of this study was to include the three most representative populations of the Chihuahua state, Northern of Mexico.

The overweight/obesity prevalence in the total studied population was 20.15%, which is lower than the national prevalence reported among Mexican teenagers in 2018 (38.4%) [4]. However, it was similar to the US national report in 2015 (20.6%) [49]. Mestizo-2 group had the highest overweight/obesity prevalence (70%) when compared to the other ethnic populations—this might be because they are from an urbanized location. Benitez et al. found that urban Tarahumara children were 10% more overweight than rural Tarahumara children in Chihuahua, Mexico [50]. The high blood pressure prevalence in the total studied population was 32.28% similar to the frequency that was reported by Salcedo-Rocha et al. (31.1%) and higher than that found by Cardoso-Saldaña et al. (19.2%) in Mexican adolescents [51,52]. The results in Mexicans are significantly higher than values reported by the NHANES 2015 in American teenagers (4.2%) [53]. In contrast, the hypertriglyceridemia and HDL prevalence in this study was lower than most reported in Mexican teenagers [54,55]. The prevalence of hyperglycemia was low (1.12%) similar to the reported by Cardoso-Saldaña et al. (1.7%), but different the reported by Camarillo-Romero et al. (4.9%) [52,56]. Among the four ethnic groups analyzed in this study, the Mennonites were the healthiest, whereas the Mestizo-2 population had the greater number of metabolic traits above normal values. The metabolic indicators we chose are considered metabolic syndrome risk factors for adults. Even though the study population appeared to be in healthy individuals—no clinical setting—there was a percentage of teenagers with values outside the expected for healthy populations. This suggests that at these early ages, some metabolic risks could be detected, thus providing an opportunity window for intervention targeted to modifiable factors of obesity and chronic non-transmissible diseases. 

PPARs play complex, overlapping, and specific roles in metabolic networks by regulating cellular energy homeostasis during lipid and carbohydrate metabolism. Therefore, PPARs have been considered as potential therapeutic targets [6]. However, the allele frequencies vary among different populations. The Pro12Ala PPAR-γ2 G minor allele frequency in the literature ranges from 9% to 25% [9,15,16,17,57,58]. In this study, the frequency in the total studied population was 17%, while other studies in Mexican population report 10–21% [11,17]. Similarly, the +294T/C PPAR-δ C minor allele frequencies vary from 12% to 31% among different populations [22,24,25,27,28,29,30,35,59,60,61]. We found that the +294T/C PPAR-δ C allele frequency in the total studied population was 12%, which is lower than what was found in other Mexican population (26%) by Rosales-Reynoso et al. [36]. Differences in both PPAR-γ2 and PPAR-δ allelic and genotype frequencies were only observed in the Tarahumara group. This might be explained by differences in ethnic admixture; however, further studies are needed.

The Pro12Ala PPAR-γ2 G minor allele has been associated with hypertension, insulin resistance, greater BMI, and lipemia profiles, mainly in adult population. With the multiple linear regression analysis, we found significant associations with several metabolic traits: systolic blood pressure, glucose, HDL, and atherogenic index. However, except for the atherogenic index, these associations were no longer significant after adjusting for ethnic group. Hasan et al. in Egyptian populations reported that the polymorphism was related to lower SBP and DBP values in patients with T2D and CHD [62]. However, most literature reports an opposite association with higher blood pressure values (SBP and DBP) in individuals from the USA, Finland, and Spain [63,64,65]. We also found a relation between the Pro12Ala PPAR-γ2 G allele and lower glucose values (adjusted by waist circumference, sex, and age). Most studies showed no association between the polymorphism and glucose levels in other populations, such as Tunisian (adjusted by age and BMI), Cypriot, Egyptian, and Emirati populations [18,58,62,66]. Other studies have reported an association of minor allele with increased insulin [15,16]. It would have been interesting to have insulin data in this study. 

In terms of lipemia, we only found an association between the Pro12Ala PPAR-γ2-minor allele and lower HDL values in the total studied population and the Mestizo-1 group (adjusted by waist circumference and sex). After the adjustment for ethnic group, this relation was no significant. In contrast, to the higher HDL levels reported in Finnish population, Becer et al. found no association between the SNP and lipemia traits in non-obese Cypriot subjects; however, obese subjects with homozygous minor genotype had increased TG values [15,58]. This suggests that obesity may interact with Pro12Ala PPAR-γ2 G minor allele and lipemia traits. As expected, we found an association between the Pro12Ala PPAR-γ2 G allele and higher atherogenic index. 

The +294T/C PPAR-δ polymorphism has been related to HDL, LDL, VLDL, and TG. With a multiple linear regression analysis, we found significant associations between the PPAR-δ polymorphism and glucose, TG, total cholesterol, HDL, and VLDL. The presence of the minor allele was associated with lower glucose values in this study (adjusted by waist circumference, sex and age). In contrast, Hu et al. found that the minor allele associated with higher glucose concentrations in Shanghai, China population (adjusted by age and sex) [67]. Whereas, no association between the C allele and glucose was observed in other Shanghai study (adjusted by age) and in Denmark subjects (adjusted by age, sex, and BMI) [30,31]. 

The +294T/C PPAR-δ C minor allele was also associated with lower TG (adjusted by waist circumference, sex and age), higher total cholesterol (adjusted by sex and age), and lower VLDL (adjusted by waist circumference, sex, and age) values in the total studied population. However, an association with lower HDL values was also found in the Tarahumara group (adjusted by waist circumference, sex, and age). This polymorphism has been associated with HDL since its discovery by Skogsberg et al. who observed its relation with lower HDL in Scottish population [24]. Similar results were reported by Aberle et al. in German woman, where +294T/C PPAR-δ-minor allele carriers had lower HDL and VLDL values (adjusted by age, smoking, and BMI) [26]. In contrast, association between the SNP and higher HDL values has been reported in Scottish and Canadian studies [27,28]. No relation was found with SBP in this study, while others have related the minor allele with lower blood pressure in Chinese population [32,68].

The logistic regression analysis with dichotomous metabolic traits values indicated that individuals with the Pro12Ala PPAR-γ2 G allele had greater risk of having an elevated waist/height ratio (adjusted by sex). Similar results were reported by Sözen et al. in Turkish populations with obesity, where the minor allele was associated with higher values of waist/height ratio; however, the same SNP was also associated with lower values in the non-obese group [69]. Whereas, Meirhaeghe et al. found no association [70]. In this study the Pro12Ala G allele carriers had two-fold less possibilities of being overweight/obese (adjusted by sex and age). In contrast to most studies in the literature, such as different meta-analysis that have shown a BMI increment in minor allele carriers [71,72]. In this study, we also found that Pro12Ala-minor allele carriers had 2.5-fold more chances of having low HDL values when adjusted by waist circumference, sex, and age and two-fold more total cholesterol (adjusted by sex and age). Regarding +294T/C PPAR-δ, we observed that C minor allele carriers had greater probability of having an overweight/obese phenotype. In contrast, this polymorphism has been related with lower BMI in Scottish males, as well as German and Chinese adult populations [27,59,73]. Nevertheless, an association was found between +294T/C minor allele and BMI in Greek toddlers [59]. These analyses were not adjusted for ethnic group, due to the low frequency in several categories. Taking into account the observed effects in the metabolic traits analyzed as continuous variables, these results may vary across the groups.

Metabolic diseases are of public health concern. For years, health programs have been focused on how to maintain normal values of clinical indicators. Unfortunately, most actions are taken in adulthood when it is already difficult to change a person’s habits. However, it should be considered that metabolic diseases depend on modifiable and non-modifiable factors. The latter are those that are related to genetic load. Ideally, the era of “Omics” aims to create networks between the different levels of information that an individual has (genome, transcriptome, metabolome, proteome, etc.). We consider that the detection of risk components for metabolic diseases at an early age would allow us to improve preventive programs for health care. To the best of our knowledge, this is the first study to report differences among ethnic populations that live in Chihuahua State from Mexico regarding PPAR-γ2 and PPAR-δ and their association/interaction with metabolic traits. Population admixtures, habits, and cultural aspects need to be considered in public health strategies. Networking disciplines, including genetics, may help to identify new therapeutic answers, and PPARs are potential therapeutic targets to control metabolic diseases. Further studies in the same ethnic populations with larger sample size and including adults will be important to confirm our findings.

## 5. Study Limitations

The cross-sectional nature of this study does not allow to follow up if participants would develop a metabolic disease in the future associated with the studied polymorphisms. The sample size by ethnic group was small for logistic regression analysis and the variable diastolic blood pressure did not reach normality with any transformation, then we used the original scale. On the other hand, the initial study was focused on early detection of metabolic syndrome components in teenagers, thus most of the studied population was in good health, contrary to most studies that include adults with specific inclusion criteria. Moreover, metabolic diseases, such as diabetes type 2, hypertension, and obesity, are multifactorial, meaning that there are external factors, such as environmental conditions, diet, and exercise, which can be modified. However, we did not explore those issues. Multifactorial diseases are also multigenic, whereas, in this study, we just included two PPAR polymorphisms and multiple SNPs analysis could still give more information. Finally, the schools were not randomly selected to assure the sample representativity. 

## 6. Conclusions

The results of this study showed that Pro12Ala PPAR-γ2 and +294T/C PPAR-δ polymorphisms are associated with metabolic indicators among ethnic populations. The total studied population showed a significant relationship between the PPAR-γ2 and PPAR-δ polymorphisms with some metabolic traits. This PPAR polymorphism study seems to be the first that includes different ethnic teenagers from the North of Mexico and, to our knowledge, it is the first report to estimate +294T/C PPAR-δ minor allele frequencies among them. An early metabolic syndrome components detection will provide the opportunity to implement prophylactic measures on modifiable factors. Further research is warranted in order to confirm the role of other PPAR polymorphisms on metabolic diseases. 

## Figures and Tables

**Figure 1 genes-11-00776-f001:**
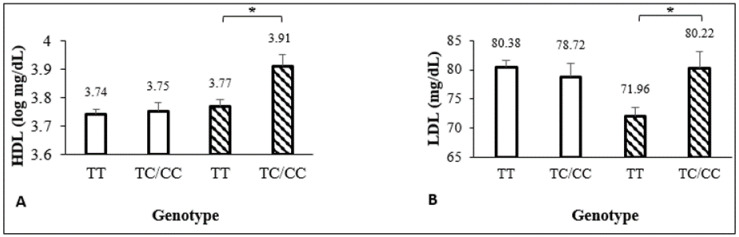
(**A**) Interaction between PPAR-δ mutated allele and sex: effect in HDL levels. *β = -0.13±0.06, *p* = 0.029. (**B**) Interaction between PPAR-δ mutated allele and sex: effect in LDL levels. *β = 9.71±4.28, *p* = 0.024. 
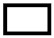
 female, 
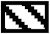
 male.

**Table 1 genes-11-00776-t001:** General characteristics of studied populations.

Characteristic	Mestizo-1 n = 96 (23%)	Mestizo-2 n = 37 (9%)	Tarahumara n = 173 (42%)	Mennonite n = 106 (26%)	Total (n = 412)
	Md (IQR)	Md (IQR)	Md (IQR)	Md (IQR)	Md (IQR)
Age (years)	13 (13–14) ^a^	13 (12–13) ^a^	14 (13–15) ^b^	15 (14–15) ^c^	14 (13–15)
Weight (kg)	47 (43–56) ^a^	59 (46–69) ^b,d^	48 (42–55) ^a^	57 (51–66) ^c,d^	51 (44–59)
Height (cm)	158 (152–164) ^a^	157 (152–162) ^a^	152 (148–158) ^b^	171 (164–175) ^c^	158 (151–166)
BMI (kg/m^2^)	19 (17–21) ^a^	24 (20–25) ^b^	20 (18–23) ^c,d^	20 (18–22) ^a,d^	20 (18–23)
	**n (%)**	**n (%)**	**n (%)**	**n (%)**	**n (%)**
Sex					
Male	41 (43)^a^	21 (57)^a^	52 (30)^b^	47 (44)^a^	161 (39)
Female	55 (57)	16 (43)	121 (70)	59 (56)	251 (61)
pBMI ≥ 85^th^ (kg/m^2^)	17 (18)^a^	26 (70)^b^	30 (17)^a^	12 (11)^a^	85 (21)

Ethnic group characteristics were compared by Kruskal-Wallis test, followed by post hoc Dunn’s test with Bonferroni adjustment ^a–d^ Different letters indicate significant differences between ethnic groups by pairwise comparisons. Abbreviations: Md = Median, IQR = Interquartile range, BMI (Body mass index, pBMI (Body mass index percentile).

**Table 2 genes-11-00776-t002:** Metabolic traits comparison among the studied populations.

Characteristic	Mestizo-1	Mestizo-2	Tarahumara	Mennonite	Total
	Mean ± SD/Md (IQR)	Mean ± SD/Md (IQR)	Mean ± SD/Md (IQR)	Mean ± SD/Md (IQR)	Mean ± SD/Md (IQR)
BMI (Z-score)	0.05 (−0.66–0.70) ^a^	1.45 (.75–1.64) ^b^	0.41 (-0.19–0.88) ^a,c^	-0.05 (-0.63–0.77) ^a,d^	0.25 (-0.50–0.92)
SBP (mmHg)	110 ± 11 ^a^	115 ± 11 ^a^	108 ± 10 ^a^	108 ± 10 ^a^	112 ± 10
DBP (mmHg)	70 (63–80) ^a^	70 (65–80) ^a,c^	70 (60–75) ^a^	75 (68–80) ^b,c^	70 (64–80)
WC (cm)	64 (61–70) ^a^	80 (72–85) ^b,e^	70 (65–75) ^c^	68 (64–74) ^d,e^	68 (63–74)
WHR	0.41 (0.38–0.44) ^a^	0.52 (0.47–0.54) ^b^	0.46 (0.43–0.49) ^c^	0.40 (0.38–0.42) ^a^	0.43 (0.40–0.48)
Glucose (mg/dL)	87 ± 10 ^a^	75 ± 6 ^b^	83 ± 11 ^a^	78 ± 9 ^c^	82 ± 10
TG (mg/dL)	77 (57–109) ^a^	93 (72–129) ^a,c^	92 (75–115) ^b,c^	75 (60–92) ^a,d^	83 (64–111)
TC (mg/dL)	145 ± 22 ^a^	156 ± 26 ^a^	133 ± 25 ^a^	144 ± 21 ^a^	140 ± 24
HDL (mg/dL)	48 (40–55) ^a^	51 (44–56) ^a^	37 (32–42) ^b^	50 (43–56) ^a^	43 (36–52)
LDL (mg/dL)	79 ± 17 ^a^	84 ± 19 ^a^	76 ± 20 ^a^	77 ± 16 ^a^	78 ± 18
VLDL (mg/dL)	15 (11–22) ^a^	19 (14–26) ^a,c^	18 (15–23) ^b,c^	15 (12–18) ^a,d^	17 (13–22)
AI (Index)	3.01 (2.59–3.48) ^a^	3.02 (2.55–3.74) ^a^	3.59 (3.12–4.20) ^b^	2.90 (2.56–3.19) ^a^	3.17 (2.76–3.69)

Ethnic group metabolic traits were compared by ANOVA or Kruskal-Wallis test for normally or non-normally distributed data, followed by post hoc Bonferroni or Dunn’s tests, respectively. ^a–e^ Different letters in the superscript indicate significant difference between ethnic groups by pairwise comparisons. Abbreviations: Md = Median, IQR = Interquartile range, SD = Standard deviation, BMI (body mass index), SBP (systolic blood pressure), DBP (diastolic blood pressure), WC (waist circumference), WHR (waist/height ratio), TC (total cholesterol), TG (triglycerides), VLDL (very density lipoproteins cholesterol), LDL (low density lipoproteins cholesterol), HDL (high low density lipoproteins cholesterol), and AI (atherogenic index).

**Table 3 genes-11-00776-t003:** Pro12Ala PPAR-γ2 genotype and allelic frequencies comparison among populations.

PPAR-γ2	Mestizo-1 n (%)	Mestizo-2 n (%)	Tarahumara n (%)	Mennonite n (%)	Total n (%)
CC	79 (82) ^a^	25 (68) ^a,c^	103 (60) ^b,c^	83 (78) ^a^	290 (70)
CG	14 (15)	10 (27)	61 (35)	20 (19)	105 (26)
GG	3 (3)	2 (5)	9 (5)	3 (3)	17 (4)
HWE *p*	0.09	0.75	1	0.45	0.18
C	172 (90) ^a^	60 (81) ^a,c^	267 (77) ^b,c^	186 (88) ^a^	685 (83)
G	20 (10)	14 (19)	79 (23)	26 (12)	139 (17)

Chi^2^ test. ^a–c^ Different letters indicate significant difference in genotype and allelic frequencies among groups *p* < 0.05. HWE *p* (Hardy-Weinberg Equilibrium *p*). G = minor allele. The Tarahumara (^b,c^) group differed from Mestizo-1 (^a^) and Mennonite (^a^), but not from Mestizo-2 (^a,c^). Mestizo-1 (^a^), Mestizo-2 (^a^) and Mennonite (^a^) were not different. The difference between Mestizo-1 and Mestizo-2 showed a *p* value = 0.06.

**Table 4 genes-11-00776-t004:** +294T/C PPAR-δ genotype and allelic frequencies comparison among populations.

PPAR-δ	Mestizo-1 n (%)	Mestizo-2 n (%)	Tarahumara n (%)	Mennonite n (%)	Total n (%)
TT	62 (65) ^a^	23 (62) ^a^	153 (88) ^b^	81 (76) ^a^	319 (77)
TC	33 (34)	13 (35)	20 (12)	23 (22)	89 (22)
CC	1 (1)	1 (3)	0 (0)	2 (2)	4 (1)
HWE *p*	0.32	0.88	0.71	0.97	0.72
T	157 (82) ^a^	59 (80) ^a^	326 (94) ^b^	185 (87) ^a^	727 (88)
C	35 (18)	15 (20)	20 (6)	27 (13)	97 (12)

Chi^2^ test. ^a,b^ Different letters indicate significant differences in genotype and allelic frequencies among groups *p* < 0.05. HWE *p* (Hardy-Weinberg Equilibrium *p*). C = minor allele. The Tarahumara group differed from the other ethnic groups.

**Table 5 genes-11-00776-t005:** Metabolic traits comparison between Pro12Ala PPAR-γ2 and +294T/C PPAR-δ genotype (major allele homozygous vs. minor allele carriers) in the total studied population.

Metabolic Traits	Pro12Ala PPAR-γ2			+294T/C PPAR-δ		
	CC (n = 290)	CG/GG (n = 122)		TT (n = 319)	TC/CC (n = 93)	
	Mean ± SD/Md (IQR)	Mean ± SD/Md (IQR)	*p*	Mean ± SD/Md (IQR)	Mean ± SD/Md (IQR)	*p*
BMI (Z-score)	0.22 (−0.54–0.99)	0.28 (-0.37–0.77)	0.46 ^#^	0.21 (-0.52–0.87)	0.39 (−0.41–1.18)	0.06 ^#^
SBP (mmHg)	111 ± 11	109 ± 10	0.08 *	110 ± 11	112 ± 11	0.15 *
DBP (mmHg)	70 (65–80)	70 (60–78)	0.10 ^#^	70 (60–80)	70 (65–78)	0.78 ^#^
WC (cm)	68 (63–75)	69 (65–74)	0.37 ^#^	68 (63–74)	69 (65–77)	0.30 ^#^
WHR	0.42 (0.39–0.48)	0.44 (0.41–0.48)	**0.04 ^#^**	0.43 (0.40–0.48)	0.42 (0.40–0.47)	0.82 ^#^
Glucose (mg/dL)	82 ± 10	82 ± 10	0.90 *	82 ± 11	80 ± 8	0.07 *
TG (mg/dL)	83 (64–112)	87 (66–110)	0.51 ^#^	87 (68–112)	75 (59–109)	**0.01 ^#^**
TC (mg/dL)	142 ± 24	138 ± 24	0.14 *	140 ± 25	143 ± 21	0.25 *
HDL (mg/dL)	45 (37–52)	39 (35–48)	**<0.01 ^#^**	43 (36–51)	48 (36–54)	**0.03 ^#^**
LDL (mg/dL)	78 ± 19	76 ± 17	0.37 *	77 ± 19	80 ± 16	0.24 *
VLDL (mg/dL)	17 (13–22)	17 (13–22)	0.51 ^#^	17 (14–22)	15 (12–22)	**0.01 ^#^**
AI (Index)	3.14 (2.70–3.66)	3.23 (2.82–3.77)	0.15 ^#^	3.18 (2.81–3.68)	3.12 (2.66–3.70)	0.37 ^#^

Groups were compared by * T-Student test or ^#^ Wilcoxon’s rank sum test for normally or non-normally distributed data, respectively. The association between PPAR-γ2 polymorphism and HDL remained significant after adjusting for multiple testing by the Holm-Bonferroni method (cut off *p* value = 0.0125) but the one of WHR did not (cut off *p* value = 0.006). The association between PPAR-*δ* and HDL did not remain significant after correcting for multiple testing; while significance remained for TG and VLDL. Abbreviations: Md = Median, IQR = Interquartile range, SD = Standard deviation, BMI (body mass index), SBP (systolic blood pressure), DBP (diastolic blood pressure), WC (waist circumference), WHR (waist/height ratio), TC (total cholesterol), TG (triglycerides), VLDL (very density lipoproteins cholesterol), LDL (low density lipoproteins cholesterol), HDL (high low density lipoproteins cholesterol), AI (atherogenic index), pBP (blood pressure percentile), and pWC (waist circumference percentile).

**Table 6 genes-11-00776-t006:** Association/Interaction between Pro12Ala PPAR-γ2 polymorphism and metabolic traits by population groups.

Metabolic Trait	Mestizo-1	Mestizo-2	Tarahumara	Mennonite	Total	Total (Adjusted)
	n = (96)	n = (37)	n = (173)	n = (106)	n = (412)	n = (412)
	β ± SE	β ± SE	β ± SE	β ± SE	β ± SE	β ± SE
	(*p*)	(*p*)	(*p*)	(*p*)	(*p*)	(*p*)
SBP (mmHg) *	**−5.59 ± 2.60**	−0.62 ± 3.03	−1.37 ± 1.44	0.61 ± 2.64	**−2.05 ± 1.07**	−1.41 ± 1.08 (0.191) *^, ++^
	**(0.035)**	(0.840)	(0.341)	(0.817)	**(0.057) ^+^**	
DBP (mmHg) *	−2.14 ± 2.34	−0.70 ± 3.36	−0.34 ± 1.37	1.25 ± 2.10	−1.68 ± 0.98	−0.73 ± 0.98 (0.455) ^‡^
	(0.363)	(0.835)	(0.802)	(0.554)	(0.086)	
Glucose (mg/dL) *	−4.77 ± 2.71	1.74 ± 2.20	0.73 ± 1.53	1.46 ± 1.81	**−18.77 ± 9.20**	−13.82 ± 8.48 (0.104) *^, --^
	(0.082)	(0.436)	(0.632)	(0.423)	**(0.042)** ^-^	
TG (log mg/dL) *	−0.02 ± 0.11	−0.07 ± 0.14	−0.03 ± 0.05	0.10 ± 0.09	0.03 ± 0.04	-0.008 ± 0.04 (0.84) ^§^
	(0.863)	(0.654)	(0.586)	(0.276)	(0.475)	
TC (mg/dL) ^#^	−4.07 ± 5.85	−10.67 ± 9.27	−1.66 ± 3.74	8.01 ± 4.57	−3.91 ± 2.59	−1.48 ± 2.53 (0.558) ^¶^
	(0.489)	(0.258)	(0.658)	(0.083)	(0.132)	
HDL (log mg/dL) ^	**−0.11 ± 0.05**	<−0.01 ± 0.08	0.03 ± 0.03	0.01 ± 0.04	**−0.06 ± 0.03**	−0.005 ± 0.02 (0.825) ^‡^
	**(0.033)**	(0.961)	(0.250)	(0.815)	**(0.031) ^~^**	
LDL (mg/dL) *	2.07 ± 4.71	−9.94 ± 6.74	−2.77 ± 2.87	4.98 ± 3.66	−1.94 ± 1.92	−9.63 ± 6.19 (0.119) ^
	(0.660)	(0.150)	(0.336)	(0.176)	(0.312)	
VLDL (log mg/dL) *	−0.02 ± 0.11	−0.07 ± 0.14	−0.03 ± 0.05	0.10 ± 0.09	0.03 ± 0.04	−0.28 ± 0.78 (0.720) ^§^
	(0.863)	(0.654)	(0.586)	(0.276)	(0.475)	
AI (log) ^	0.08 ± 0.05	−0.07 ± 0.07	−0.05 ± 0.03	0.05 ± 0.04	**0.40 ± 0.19**	**0.39 ± 0.17 (0.021) ** ^^,^ ^//^
	(0.114)	(0.367)	(0.084)	(0.244)	**(0.033) ^/^**	

Analyses on total studied population were adjusted for waist circumference and/or sex, and/or age, as indicated in variable names and ethnic groups. Total studied population adjusted models included ethnic groups as fixed effects, in addition to those indicated by the superscript. Variables with non-normal distribution were log transformed (HDL, TG, VLDL and AI). Coefficients for interaction terms are described below and indicated by symbols in the table. * Adjusted for waist circumference, sex and age. ^#^ adjusted for sex and age. ^ adjusted for waist circumference and sex. ^‡^ adjusted for waist circumference. ^§^ adjusted for waist circumference and age. ^¶^ adjusted for sex. ^+^ Interaction sex X age (β = −2.10 ± 0.72, *p* = 0.004), ^++^ Interaction sex X age (β = 1.84 ± 0.72, *p* = 0.011). ^-^ interaction polymorphism X waist circumference (β = 0.27 ± 0.13, *p* = 0.040), ^--^ interaction polymorphism X waist circumference (β = 0.20 ±0.12, *p* = 0.10). ^~^ interaction sex X waist circumference (β = 0.007 ± 0.003, *p* = 0.008). ^/^ interaction polymorphism X waist circumference (β = −0.005 ± 0.003, *p* = 0.047), ^//^ interaction polymorphism X waist circumference (β = −0.006 ± 0.002, *p* = 0.019). Abbreviations: SBP (systolic blood pressure), DBP (diastolic blood pressure), TC (total cholesterol), TG (triglycerides), HDL (high density lipoproteins cholesterol), LDL (low density lipoproteins cholesterol), VLDL (very low-density lipoproteins cholesterol), and AI (atherogenic index).

**Table 7 genes-11-00776-t007:** Association between +294T/C PPAR-δ polymorphism and metabolic traits in studied populations.

Metabolic Trait	Mestizo-1	Mestizo-2	Tarahumara	Mennonite	Total	Total (Adjusted)
	n = (96)	n = (37)	n = (173)	n = (106)	n = (412)	n = (412)
	β ± SE	β ± SE	β ± SE	β ± SE	β ± SE	β ± SE
	(*p*)	(*p*)	(*p*)	(*p*)	(*p*)	(*p*)
SBP (mmHg) *	2.07 ± 2.09	0.74 ± 2.97	1.71 ± 2.26	−1.54 ± 2.55	1.94 ± 1.19	1.34 ± 1.20
	(0.326)	(0.806)	(0.451)	(0.546)	(0.103)	(0.263) *
DBP (mmHg) ^#^	−0.25 ± 1.84	−0.38 ± 3.25	0.29 ± 2.10	−0.67 ± 2.03	0.85 ± 1.08	0.01 ± 1.08
	(0.894)	(0.907)	(0.889)	(0.742)	(0.432)	(0.994) ^#^
Glucose (mg/dL) *	−2.62 ± 2.15	0.04 ± 2.17	−3.91 ± 2.39	−0.75 ± 1.75	**−2.57 ± 1.18**	**−2.22 ± 1.12** **(0.048) * **
	(0.227)	(0.984)	(0.103)	(0.669)	**(0.031)**	
TG (log mg/dL) *	−0.10 ± 0.09	−0.12 ± 0.14	−0.07 ± 0.08	−0.15 ± 0.09	**−0.12 ± 0.05 **	−0.08 ± 0.05 (0.08) ^#^
	(0.273)	(0.415)	(0.374)	(0.086)	**(0.008)**	
TC (mg/dL) ^	−0.002 ± 4.60	2.12 ± 9.35	−4.89 ± 5.87	−5.76 ± 4.48	**79.02 ± 29.61**	49.8 ± 28.9 (0.085) ^&, ++^
	(1)	(0.822)	(0.406)	(0.202)	**(0.008) ^+^**	
HDL (log mg/dL) *	0.02 ± 0.04	0.11 ± 0.07	**−0.10 ± 0.05**	0.01 ± 0.04	<0.01 ± 0.04	0.08 ± 0.07 (0.256) ^‡^
	(0.594)	(0.144)	**(0.046)**	(0.685)	(0.904) ^-^	
LDL (mg/dL) *	−6.75 ± 4.64	2.66 ± 6.6	0.47 ± 4.52	−4.15 ± 3.49	−1.67 ± 2.66	−2.89 ± 2.68 (0.281) ^‡,~~^
	(0.149) ^/^	(0.688)	(0.917) ^	(0.238) ^&^	(0.531) ^‡,~^	
VLDL (log mg/dL) *	0.10 ± 0.09	−0.12 ± 0.14	−0.07 ± 0.08	−0.15 ± 0.09	−0.12 ± 0.05	−0.08 ± 0.05 (0.082) ^#^
	(0.273)	(0.415)	(0.374)	(0.086)	(0.008)	
AI (log) *	−0.02 ± 0.04	−0.08 ± 0.07	0.08 ± 0.05	−0.06 ± 0.04	−0.03 ± 0.02	−0.03 ± 0.06 (0.599) ^#^
	(0.643)	(0.223)	(0.093)	(0.133)	(0.208)	

Analyses on total studied population were adjusted for waist circumference and/or sex, and/or age, as indicated in variable names and ethnic groups. Total studied population. adjusted models included ethnic groups as fixed effects, in addition to those indicated by the superscript. Variables with non-normal distribution were log transformed (HDL, TG, VLDL and AI). Coefficients for interaction terms are described below and indicated by symbols in the table. * Adjusted for waist circumference, sex, and age. ^#^ adjusted for waist circumference and age. ^ adjusted for sex and age. ^&^ Adjusted for age. ^‡^ Adjusted for waist circumference. ^+^ Interaction polymorphism X age (β = −5.53 ± 2.14, *p* = 0.010), ^++^ Interaction polymorphism X age (β = −3.69 ± 2.08, *p* = 0.076). ^-^ interaction polymorphism X sex (β = −0.13 ± 0.06, *p* = 0.029). ^~^ Interaction polymorphism X sex (β = −9.71 ± 4.28 *p* = 0.024), ^~~^ Interaction polymorphism X sex (β = −8.99 ± 4.27 *p* = 0.035). / interaction polymorphism X and sex (β = −18.61 ± 7.27 *p* = 0.012). Significant differences in bold *p* < 0.05. Abbreviations: SBP (systolic blood pressure), DBP (diastolic blood pressure), TC (total cholesterol), TG (triglycerides), HDL (high density lipoproteins cholesterol), LDL (low density lipoproteins cholesterol), VLDL (very low density lipoproteins cholesterol), and AI (atherogenic index).

**Table 8 genes-11-00776-t008:** Association between Pro12Ala PPAR-γ2 or +294T/C PPAR-δ with metabolic traits in the total studied population.

Metabolic Traits	Pro12Ala PPAR-γ2				+294T/C PPAR-δ			
	CC	CG/GG	Adjusted OR (_95%_CI)	*p*	TT	C/CC	Adjusted OR (_95%_CI)	*p*
	(n = 290)	(n = 122)			(n = 319)	(n = 93)		
	n (%)	n (%)			n (%)	n (%)		
BMI (Z-score > 1.04)	68 (23)	15 (12)	**0.46 (0.25–0.84)**	**0.011**	57 (18)	26 (28)	**1.78 (1.04–3.05)**	**0.034**
pBP ≥ 90th	100 (34)	33 (27)	0.70 (0.44–1.12)	0.142	103 (32)	30 (32)	1 (0.61–1.64)	0.996
pWC ≥ 90th	9 (3)	5 (4)	1.33 (0.44–4.07)	0.612	9 (3)	5 (5)	1.96 (0.64–5.99)	0.239
WHR > 0.45 *	108 (37)	55 (45)	**1.79 (1.03–3.08)**	**0.036 ^¶^**	131 (41)	32 (34)	0.74 (0.45–1.21)	0.231
Glucose ≥ 110 (mg/dL) **	4 (1)	1 (1)	--		5 (2)	0 (0)	--	
TG ≥ 110 ^ (mg/dL)	76 (26)	30 (25)	0.91 (0.56–1.50)	0.721	84 (26)	22 (24)	0.77 (0.44–1.34)	0.350
TC ≥ 150 ^#^ (mg/dL)	41 (14)	9 (7)	**0.47 (0.22–1)**	**0.050 ^-^**	39 (12)	11 (12)	0.87 (0.43–1.81)	0.726 ^~^
HDL ≤ 40 ^@^ (mg/dL)	92 (32)	65 (53)	**2.50 (1.60–3.89)**	**<0.01**	128 (40)	29 (31)	1 (0.55–1.82)	0.995 ^/^
LDL ≥ 110 ^‡^ (mg/dL)	13 (4)	3 (2)	0.54 (0.15–1.96)	0.351	15 (5)	1 (1)	0.20 (0.03–1.52)	0.118
VLDL ≥ 30 ^‡^ (mg/dL)	22 (8)	7 (6)	0.74 (0.31–1.80)	0.514	23 (7)	6 (6)	0.82 (0.32–2.10)	0.679
AI (M > 4 F > 3.5) ^&^	86 (30)	39 (32)	1.07 (0.64–1.78)	0.798	99 (31)	26 (28)	0.80 (0.45–1.41)	0.435

Reference: homozygous group for major allele. * Adjusted by sex,^#^ adjusted by sex and age, ^ adjusted by waist circumference and age, ^@^ adjusted by waist circumference, sex and age, ^‡^ adjusted by waist circumference, ^&^ adjusted by sex and WC. ^¶^ Interaction polymorphism X sex (β = 0.46 ± 0.22, *p* = 0.108), ^-^ interaction sex X age (β = 0.39 ± 0.14, *p* = 0.009), ^~^ interaction sex X age (β = 0.40 ± 0.14, *p* = 0.010), / interaction polymorphism X sex (β = 0.27 ± 0.17, *p* = 0.040). ** Odds ratio was not calculated, due to the low frequency of individuals in the high-glucose category. Abbreviations: pBP (blood pressure percentile), pWC (waist circumference percentile), WHR (waist-height ratio), TC (total cholesterol), TG (triglycerides), HDL (high low density lipoproteins cholesterol), LDL (low density lipoproteins cholesterol), VLDL (very low density lipoproteins cholesterol), and AI (atherogenic index) M = male, F = female.

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
