# Peer review of "Pro12Ala PPAR-γ2 and +294T/C PPAR-δ Polymorphisms and Association with Metabolic Traits in Teenagers from Northern Mexico"

_genes, 2020, doi:10.3390/genes11070776_

Round 1
Reviewer 1 Report
The authors present a thorough characterization of PPAR-gamma and -delta polymorphisms and their associations with complex metabolic traits within a cohort of Mexican teenagers from different ancestries. The scientific methodology employed in this study is sound and the gleaned insights concern clinical relevant polymorphisms in an epidemiologically important, yet understudied, ancestral group, providing further value in gained knowledge. However, the manuscript is unfocused in sections as it details an extensive breadth of results, without clearly contextualizing the meaningfulness of the results. Moreover, the manuscript details a number of associations with small effect sizes that barely meet statistical significance (in some cases with p-values > 0.05). It would be better to reduce the emphases on such associations and instead expand upon the meaningfulness of the more robust associations.
Below I've included my comments and suggestions for improving the quality of the manuscript:
Line 31/33: Not clear which allele is the "mutated" allele for this polymorphism.
Line 33: Not clear what specific "overweight/obesity" phenotype is being measured. BMI?
Line 33-34: Not clear what "risk factor high Z-score values" actually means. Aren't Z-score inherent to the association test per se?
Line 34: Not clear what metabolic traits are "abnormal". Please consider revising.
Line 47: I wonder if "derived" would be a more appropriate term than "mutated" in this context as you are contrasting an ancestral allele with one that was introduced at a later point in history, but that is presumably segregating at an appreciable frequency in the population?
Line 48: What specific allele has been associated with increased risk? C or G? Or does it vary with the trait?
Line 60: Why use the full base names here but not in the previous paragraph?
Line 74: I'm a bit confused about what the "adult" overweight/obesity prevalence means when the cohort is 12-19 year olds?
Line 76-77: Citations?
Introduction: I wonder if there was a reason why this investigation focused on teenagers in particular? Are there specific epidemiological or biological justifications for investigating this demographic in particular? If so, I think it would be include the justifications in the introduction.
Lines 89-91: What was the composition and sampling scheme for the "mixed" group?
Line 112: Remove "like".
Table 1: "Media" typo under Mennonite column.
Table 1: I am a bit confused by values Table 1. The column units include Mean +/- SD, but all values look like Md (IQR) to me. Also why is there mention of KW and ANOVA tests in the table footnotes? I don't see any test results here.
Line 164: What is AI here? It doesn't seem to be defined in the introduction, or Methods for that matter.
Line 165: Same with DBP. I assume this is diastolic blood pressure, but not defined anywhere. Also, the amount of acronyms in this paragraph makes this section hard to follow.
Line 172: What does the 31.34% actually refer to?
Results section 3.1: Overall, this section was hard to follow as its cumbersome to keep track of the numereous acronyms and percentages. Would be better to highlight to most pronoucned differences between groups and expand on them.
Table 2: I clearly follow results from statisical comparison tests in this table; or its not clear how I am supposed to read the a,b,c,d results in description. Also, not clear which traits are normally and non-normally distributed.
Line 177: Genotype information was not included in Table 2. Perhaps you mean Table 3 here?
Line 178: Table 3 seems to be mislabeled as well.
Line 190: Why was Z-score BMI compared rather than BMI in kb/m^2?
Table 5: Given that you are evaulating multiple traits, do the HDL associations remain significant when you correct for multiple testing (i.e. Bonferoni adjustment) to control the family-wise error rate?
Lines 202-203: Not sure what to make of this when the Pro12Ala genotype showed no significant relationships with glucose or WC measurement in Table 5.
Lines 209-210: Would these associations be significant if you adjusted for population as a fixed effect?
Line 234: Not sure if "possibilities" is the right word here.
Line 249: Perhaps "total sample" would be better than "total population" in this context?
Line 263: It wasn't clear in what sense the Mixed population was "mixed". Similarly, what does it mean to have "more altered metabolic traits"?
The discussion is rather long. Some of the epidemiological information could have instead been incorporating into the introduction. Also, the discussion retreads much of the results already described without providing much further insight. For example, the allele frequencies, within each ethnic group, are restated and shown to differ with those characterized in other studies, but no much insight is given into why these differences may exist.
Perhaps, rather than listing all the contrasting associations between these variants and metabolic traits reported by previous studies in the discussion, it would be better to collate cross-study comparisons into a table to better contrast the effects observed in the current study? This may be instead be incorporated in the results as a formal analysis of the consistency of the observed effects across studies/ancestral populations?
Author Response
"Please see the attachment."

Reviewer 2 Report
It is an interesting study by Venzor et al., on the effect of Pro12Ala and +294T/C polymorphisms on metabolic traits in teenagers from Northern Mexico, where the authors also addressed the differences in the traits between different ethnic groups. Although it is a well-designed study with interesting findings, the authors need to correct for certain grammatical mistakes and answer the concerns mentioned below:
Major Comments:
1) Cut short the introduction. The introduction should start with the prevalence of obesity/ metabolic diseases and traits instead of a description about the gene, as the study is about the genetic basis of abnormal metabolic traits.
2) Line no. 76 “A few studies in Mexican teenagers have shown association with Pro12Ala PPAR-γ276 polymorphism and obesity-related traits” is contradictory to line no. 79, “No association was seen between 78 +294T/C PPAR-δ polymorphism and metabolic traits in Mexican children”. Please correct it.
3) Define the mixed population as it is not clear.
4) How was WC p<90 decided? Cite the paper if it has been taken from literature.
5) ANOVA and Kruskal -Wallis test are very similar. Why were both methods used? Please clarify in the manuscript methods under statistical analysis.
6) Use risk allele instead of mutated allele, as the authors are studying a polymorphism and not a mutation
7) Which all variables were log transformed due to non-normal distribution?
8) Please clarify how logistic regression was used for association analyses in the method section. Which of the phenotypes were continuous or categorical.
9) Use n (%) for population size and gender in table 1.
10) Explain a,b,c,d in the legend in detail for tables as it is hard to understand.
11) Which all clinical covariates were significantly associated with the phenotype? Please provide the p value. Were those significantly associated covariates adjusted for in the genetic association analyses?
12) The authors observed significant allele frequency difference between different ethnicities. Wouldn’t that bias the genetic association analyses for the total population?
13) For genetic association analyses, data should be normally distributed to avoid type I error. If normality does not hold, a transformation (for example, log) of the original trait values might lead to approximate normality.
14) Line 230, “approximately 80% greater likelihood of having an elevated WHR” how did the authors achieve this number -80%. All numbers in the text should match with the numbers in the table.
15) Reduce the discussion sufficiently and concentrate only on the main aspects and findings of the paper.
16) Conclusion needs to be reduced sufficiently. Please add a section on study limitations
Minor Comments:
- Remove the sentence on line n. 42, “and seven mRNA result of alternative splicing” as this is not true. Correct it, refer and cite PMID 18435931
- Correct grammatical mistakes:
a) Line 44, change to “specific to adipose” - b) Line 175, change to no ‘significant’ difference
3) Change PPARG2 to PPARG when talking about the gene or polymorphism.
4) Line 133, change to 269 bp
5) Line 177, change to Table 3
6) Line 178, change to Table 4
7) Line 180, remove the Table 3 and 4.
8) Line 178-180, Reform the sentence as “the polymorphisms were in HWE in all the ethnic groups and total population”
Author Response
"Please see the attachment."

Round 2
Reviewer 1 Report
The reviewer greatly appreciates the efforts made by the authors in addressing to comments put forward. Overall, the reviewer is satisfied with the scientific soundness of the manuscript and the improvements conferred through the additional analyses performed by the authors. The authors provided clarity on key concerns of the reviewer and enhanced the clarity of the writing.
The reviewer would still appreciate clarity on the point concerning the "A,B,C,D" superscript denotation used in Table 1 and Table 2. It is understood that "[D]ifferent letters indicate significant differences between groups p <0.05" but it is not clear to the reviewer if the letters indicate degrees of significance or specific groups. Do the groups correspond to the columns? How should one interpret a number with the superscript "a,d"? This scheme is unfamiliar to the reviewer and further clarification would be greatly appreciated. Also, Table 2 includes a "e" superscript as well which has not been addressed or explained in the table footnote.
Aside from this, there are a few minor revision comments that are offered as suggestions to improve the clarity of the writing further:
Line 27: Perhaps say ancestral and derived; OR major / minor?
End of Introduction:
I would suggest moving the sentence on lines 78-81 ("The reason ....") to the end of the paragraph. Also, you can remove "The reason" and begin with "We focused..". I'd also revise the sentence a bit further as the grammar is a bit awkward and retain the latter content on prevention. Otherwise, I am satisified with the correction.
Line 92: Replace "a different geographic location" with "different geographic locations".
Table 2 footnote: Double period: "groups.."
Line 282: Replace "North of Mexico" with "in Northern Mexico".
Line 299: Remove extra hyphen.
Line 318: Add period before "Hasan et al."
Reviewer 2 Report
The authors have made significant improvement in the manuscript. However there are still some concerns that need to be addressed and corrected by the authors.
1) Based on Line 156, it seems like the authors used non-normally distributed data for genetic association analysis, which can lead to bias in the results. Table 5 also looks like non normally distributed data was used for genetic association analyses. If the authors used log transformed values, then please re-frame the sentences and clarify to avoid confusion/errors.
2) It is not clear how the authors used Bonferroni correction for multiple testing? Line 213 is confusing.
3) Did the authors use covariates for the analyses shown in table 5? If not, please use the covariates and provide adjusted p values in this table.
4) There are too many tables in the main manuscript. Advise to move table 6 and 7 to supplementary. Please represent the tables similar to table 5 but for each population.
5) Not sure what the details with superscript ‘+’ means on Table 8. Also the superscript doesn’t appear anywhere in the table.
6) Write a sentence after line 47 on the factors associated with obesity and the influence of genetics. This will help in connecting the first two paragraphs of the introduction. Include a line or two on PPAR role in lipid and glucose metabolism, and overall energy homeostasis.
7) Line 55: Mention, which is the major and minor allele
8) Either use major/ minor or ancestral/derived
9) Italicize et.al, throughout the manuscript
10) Line 75: Change to “shown association of Pro12Ala PPAR-γ2 polymorphism with obesity-related traits.”
11) Line 95: Change to ‘haplotype’ frequencies
12) Move line 163 before line 156 if this is true.
13) Change total sample to total population throughout the manuscript.
14) Define a, b, c, d clearly as what each means for the tables 1, 2 and 3.
15) Line 211: Change to WHR.
16) Please move the bottom part of table 5 [n(%)] to table 8.
